

# Evaluation of the effects of different lightning protection rods on the data quality of C-Band weather radars

Cornelius Hald[1], Maximilian Schaper[1], Annette Böhm[1], Michael Frech[1], Jan Petersen[2], Bertram Lange[2], and Benjamin Rohrdantz[2]

[1]Observatorium Hohenpeißenberg, Forschung und Entwicklung, Deutscher Wetterdienst, Albin-Schwaiger-Weg 10, 82383 Hohenpeißenberg, Germany
[2]Technische Infrastruktur und Betrieb, Deutscher Wetterdienst, Frahmredder 95, 22393 Hamburg, Germany

**Correspondence:** Cornelius Hald (cornelius.hald@dwd.de)

**Abstract.** Lightning protection is important for weather radars to prevent critical damage or outages but can have negative effects on data quality. The existing lightning protection of the DWD polarimetric C-Band weather radar network consists of four vertical poles with a maximum diameter of 10 cm. During radar operation, these rods cause local scattering in the near field of the antenna, resulting in negative impacts on radar products. One effect is the removal of significant transmit power

off the main beam axis and its addition to other areas or the side lobes. This results in wrongly localised precipitation fields in radial direction. The second effect is the loss of transmitted and received power, appearing as a decrease in system gain, and subsequently a underestimation of all power based radar moments in the vicinity of the rods. The underestimation in radar reflectivity Z then leads to a negative bias in the actual rain rate of approximately 20% if a Z/R relationship is applied. These detrimental effects on data quality led to the requirement of developing a new lightning protection concept. The new concept

must minimise the effect on data quality, but also provide sufficient protection from lightning strikes according to the existing regulations and requirements. Three possible lighting protection concepts are described in this paper: two using vertical rods of different diameters (16 and 40 mm, respectively) and one with horizontally placed rods outside the antenna aperture. Their possible influence on data quality is quantified through a dedicated measurement campaign by analysing resulting antenna patterns and precipitation sum products. Antenna patterns are analysed with respect to the side lobe levels compared to antenna

patterns without lightning protection and the original lightning protection. With the newly tested lightning rods, the apparent side lobe levels are slightly increased compared to an antenna pattern taken without lightning protection, but are within the accepted antenna specifications. Compared to the original lightning protection, a decrease of up to -15 dB in apparent side lobe levels is found for all tested lightning protection options. Beam blockage is substantially reduced compared to the existing lightning protection as shown by the evaluation of QPE sums. These results and some structural considerations are a solid basis

to recommend the installation of four rods with maximum 40 mm diameter for all 17 radar systems of the DWD weather radar network.





# 1 Introduction

Weather radars are often located on towers in very exposed locations to have an unobstructed view around the entire azimuth (AZ). This can increase the risk of lightning striking the devices (Kingfield et al., 2017). To prevent this, the radar installations

are usually secured by an external lightning protection installation to avoid damage to the radar or radome and guarantee the safety of radar technicians. Lightning protection is commonly realised by having a number of lightning protection rods (LPR) around or on top of the radome that are able to conduct electric energy of a lightning strike from the top of the tower to the ground.

A common concept consists of four or more vertical LPRs around the radome, sometimes complemented with a LPR on

top of the radome. Such a setup means that the rods reach into the aperture of the antenna and can therefore interact with the electromagnetic fields during transmit and receive. Scattering of electromagnetic waves on the LPRs cause two negative effects on data quality: The first is the emission of a part of the transmitted pulse outside of the main lobe of the antenna. In comparison to an antenna without LPRs, this phenomenon manifests in the antenna pattern by way of increased side lobe levels. In an actual radar image of a thunderstorm this appears as an echo that resembles a so-called hail-spike (see Fig. 3 and

Zrnic et al., 2010). For the remainder of this paper, we will be referring to this phenomenon as reflection. The second effect of the parasitic scattering is a reduction in the main lobe energy, which leads to an underestimation of precipitation in the effected direction. It appears as shadowing effects in accumulated weather radar measurements and is therefore often termed beam blockage in the literature (e.g. Krajewski et al., 2006; McRoberts and Nielsen-Gammon, 2017).

The German Meteorological Service (Deutscher Wetterdienst, DWD) is operating 17 operational and one research C-Band

weather radars (Frech et al., 2017). They have been in service since 2012. The lighting protection concept consists of four LPRs with a maximum diameter of 10 cm and a length of 8 m. These LPRs are known to show both of the negative effects on data quality described above. It was therefore decided to introduce a new concept with less influence on data quality while keeping the required level of lightning protection to meet all safety concerns. Three new types of LPRs were evaluated: Two vertical rods with a length of 4 m and a diameter of 16 mm and 40 mm, respectively, and a horizontal rod with a length of 4 m and a

diameter of 76 mm. All three options require an additional LPR of about 1 m length in the top panel of the radome, which is not part of this evaluation. Previous investigations looking at Doppler spectra recorded at 90° elevation found no influence on the radar's operation.

The effect of the three LPR options on data quality is evaluated through antenna pattern measurements and dedicated data analysis. To this end, a measurement campaign was conducted at the research radar of the DWD at the meteorological obser-

vatory Hohenpeissenberg (MOHP). The recorded data are compared to measurements with the old LPR and without any LPR, taken during the acceptance test of the research radar in 2010 and 2011 (see e.g. Frech et al. (2013) for some of the results). The following section (Section 2) contains an in-depth description of the three new LPR configurations and the old configuration as well as a description of the conducted measurements. Section 3 describes the results of the antenna measurements and the precipitation sums. The paper concludes with a discussion of the results and a recommendation for a future lightning protection

concept in Section 4.



## 2 LPR setups and measurements

### 2.1 Description of the lightning protection rods

The current lightning protection for the operational German weather radars is realised by four rods that are positioned on top of the radar tower surrounding the radome. These rods have a total length of 8 m. The last metre is made out of stainless steel and has a diameter of 10 mm. It is connected to an aluminium cable with a diameter of 8 mm that reaches down to the mount of the rod and is encased by fibreglass. The coating narrows down from the mounting to the stainless steel rod from 100 mm to 50 mm. The four rods are separated by 90° in azimuth and their exact distance from the antenna or the center of the tower depends on the respective tower itself, as they are all unique buildings. During the recent measurements presented here, three of the original LPR were removed, keeping only one at an azimuth of 357°. When a new radome was installed in 2011, detailed antenna pattern measurements with the old LPR at 276.5° AZ were carried out (Frech et al., 2013), which are used as a reference for comparison in the study at hand.

Of the three evaluated new LPRs, two are vertically mounted with a total length of 4 m. The thinner rod starts with a diameter of 16 mm up to a height of 3 m and narrows down to a diameter of 10 mm in the uppermost metre. The thicker rod starts with 40 mm and continues with 16 mm and 10 mm in the third and fourth metre, respectively. Both are made from stainless steel tubes and have no coating. For a complete lightning protection in accordance with the regulations, four of these are required with a spacing of 90°, together with a single shorter rod on top of the radome. During the measurements, both were placed at 267.5° AZ (same location as one of the old LPRs) and at 250° AZ. For the precipitation sums, the 16 mm rod was also placed at 177.5° AZ for a few weeks. Figure 1 shows the 16 mm LPR installed at 267.5° AZ.

The third tested LPR also has a total length of 4 m, but the first 3 m are installed horizontally, reaching outwards radially from the radar tower. The last metre is angled upwards by 45°. The diameters are 76 mm and 10 mm for the first and second part, respectively. Depending on the desired level of lightning protection, up to eight of these rods are required, together with the shorter top rod. The number of rods can also be lowered by increasing the length of the horizontal part, while retaining the same level of lightning protection. Due to it requiring a different mounting than the other rods, the horizontal LPR was only installed at 245° AZ. The horizontal LPR is shown in Fig. 2. The properties of all LPRs used in this study are summarised in Table 1.

All tested LPR options are chosen in accordance to the legal requirements of lightning protection class 3 for the MOHP radar tower. The classes can be assessed using the rolling sphere method with a maximal sphere diameter of 45 m. Thus, a shorter top panel lightning rod is a necessity.

### 2.2 Antenna measurements

Antenna measurements can be used to precisely describe the properties of the propagation path of an antenna (Chandrasekar and Keeler, 1993). For weather radars, the overall antenna pattern is of particular interest, which can be characterised by the main lobe (half-power beam width) and additional side lobe sensitivity. Furthermore, the influence of the struts and feed horn

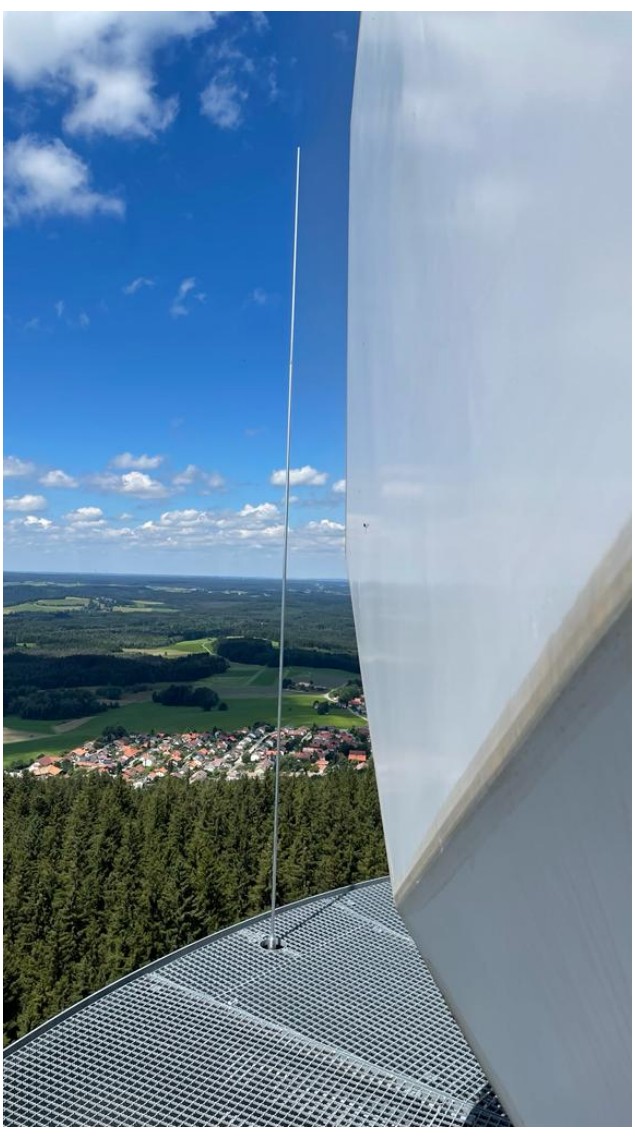

**Figure 1.** 16 mm vertical LPR installed at 267.5° AZ at the radar MOHP during the measurement campaign.

on the transmitted electrical field can be identified in the antenna pattern. For dual-pol weather radars, a good match of the characteristics between the horizontal and the vertical channel is another factor of interest.

To conduct an antenna measurement, a high-quality, stable external signal source is set up at a suitable location (in direct line of sight) in the far field of the radar. It transmits a continuous wave signal tuned to the radar frequency towards the radar. The weather radar is set into receiving mode with the transmitter turned off. A series of high resolution scans around the position of the external source is then carried out. During recent years, the use of UAVs as external signal source has become more prominent (Umeyama et al., 2020).



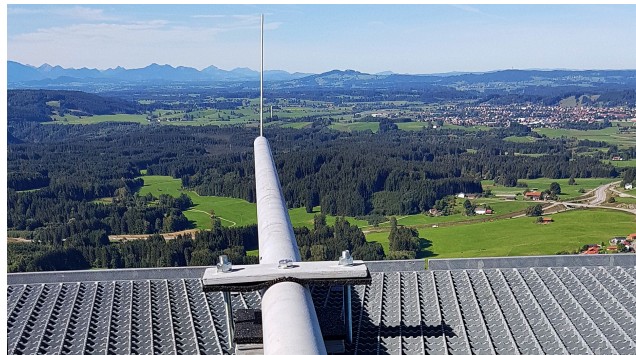

**Figure 2.** Horizontal LPR installed at 245° AZ at the Radar MOHP during the measurement campaign.

**Table 1.** Properties of the LPRs used in the presented study

| Designation | orientation | length | diameter (w/ respect to length) | position(s) |
|---|---|---|---|---|
| old LPR | vertical | 8 m | 100 mm to 50 mm (0-7 m), 10 mm (7-8 m) | 267.5°, 357° |
| 16 mm LPR | vertical | 4 m | 16 mm (0-3 m), 10 mm (3-4 m) | 250°, 267.5° |
| 40 mm LPR | vertical | 4 m | 40 mm (0-2 m), 16 mm (2-3 m), 10 mm (3-4 m) | 177.5°, 250°, 267.5° |
| horizontal LPR | horizontal, 45° upwards | 4 m | 76 mm (0-3 m), 10 mm (3-4 m) | 245° |

In the case presented here, the external signal source consists of a signal generator, a power amplifier, and a reflector antenna with a diameter of 1.3 m with a center-fed single-pol C-band feed horn. During the measurements the source unit was located on a hill called Auerberg. It is visible as the solitary hill in the foreground to the right of the LPR in Fig. 2. The orography (valley between Auerberg and MOHP) ensures an unobstructed propagation path. The distance to the radar MOHP is 21.6 km and the angles relative to the radar are 250° AZ and 0° elevation (EL). During the measurements for the old LPR in 2012, the

external signal source was installed at the same location. This setup represents a very good antenna test range, as the antenna manufacturer's pattern measurements can easily be reproduced (Frech et al., 2013).

    To ensure that the external signal source is aligned directly towards the weather radar, the following procedure is performed:

– The radar antenna is pointed towards the approximate direction of the external source. First guess coordinates can be derived from a map and elevation data.

– The radar transmitter is turned off and a pulse repetition frequency (PRF) of 3000 Hz is chosen in order to achieve a high sampling rate in the receiver.

– The received signal to noise ratio (SNR) over range is displayed live via the radar software



- The antenna of the external source is tilted carefully horizontally and vertically until the maximum signal strength is reached at the weather radar.

- The feed horn at the external site is rotated until either the signal is identical in both polarisation channels (dual polarisation mode) or until one is at maximum while the other is at minimum (single pol measurement). It is usually easier to identify the minimum.

- The external signal source antenna is fixed in place when the desired signal strength is reached.

- The radar antenna is moved in small steps around the previously used coordinates, again until a maximum SNR is
reached in either one or both channels.

- The now precisely determined azimuth and elevation angles of the signal source are recorded for use in the evaluation of the data. Here, the identified precise angles are 250.43° AZ and 0.13° EL.

It is recommended to keep contact with the operators of the external source via phone during this process or alteratively have remote control over either the external signal source or the weather radar. During the setup and the subsequent scanning, the
automatic frequency control (AFC) has to be disabled and the radar receive frequency is set manually to the transmit frequency of the external signal source.

The MOHP weather radar is a DWSR-5001C/SDP/CE by EEC with radar operating software and signal processing by Gamic. Its soft- and hardware enable it to perform a multitude of different scan types automatically and in sequence. A schedule of scans was set up prior to the measurements and the precise location was inserted into the scan templates according
the orientation described above. This way, several measurements were conducted in sequence ensuring comparable results.

All scans record data up to a range of 20 km, while 40 data bins are averaged in the signal processor to 1 km. This is done only for having a range of data to perform statistical evaluations; the total recorded range is not important, since the external source is providing a constant continuous wave and not pulsed signal. It is therefore constant in all recorded ranges. The PRF of the radar is set to is at 3000 Hz and the pulse width to to 0.4 μs. Recorded radar moments are SNR in the horizontal and
vertical channel, the differential reflectivity ZDR, differential phase PhiDP and the cross-correlation coefficient RhoHV, all from the time series without any clutter correction or thresholding. All scans were performed with an antenna speed of $6° \, \text{s}^{-1}$, resulting in about 25 pulses per ray and an angular resolution of 0.05°. Some scans utilised the time sampling mode, where the amount of pulses per ray is defined by the PRF and the recording time. One complete scan cycle takes a little over 2 hours. Several cycles with different setups of LPRs were performed; the data from the following LPR configurations will be shown in
this paper:

- no lightning protection rod, dual-pol mode

- 16 mm LPR at 267.5° AZ, dual-pol mode

- 40 mm LPR at 267.5° AZ, dual-pol mode





— horizontal LPR at 245° AZ, dual-pol mode

One cycle consists of several different types of scans as explained in the following subsections.

### 2.2.1 Source raster scan

The first scan is a raster scan that scans a rectangular area around the external signal source. It is used to repeatedly determine the precise signal strength and location of the external signal source during the scanning schedule. Both can change minimally due to changing beam propagation conditions caused by the temporal variability of pressure, temperature and humidity in the

atmosphere (Grabner and Kvicera, 2011) or because of drift in frequency or transmitted power of the external signal source. The source raster scan covers an area of ±2° in AZ and EL around the previously precisely determined position of the signal source. The horizontal resolution is set to 0.05°, the vertical resolution is limited by the software to 0.1°. To reach identical resolutions in AZ and EL, the scan is performed twice with an offset in the starting EL angle of 0.05°. Therefore, the first part goes from -2° to 2° EL and the second from -1.95° to 2.05° EL. The two scans are combined into one data set in post-processing.

### 150 2.2.2 Large raster scan

The large raster scan covers an area of ±45° in AZ around the external signal source and reaches from -2° to 20° in EL. The recorded data provides a two-dimensional image of the characteristics of the main lobe and multiple side lobes. It can be used to describe the influence of the lower struts in detail. To find a compromise between precision and duration of the measurements, the resolution from -2° to 3.05° EL is set to 0.05° (by using the offset in the starting elevation described above), to 0.1° from

3° to 9.9° EL and to 0.5° from 10° to 20° EL. The full area is split into 15 unique sweeps over the whole AZ-range and with different EL-ranges chosen such that they all take approximately the same time. After every slice, one source raster scan is performed that is later used to correct for signal strength and location. The slices are again put together to one large data set in post-processing.

### 2.2.3 Plan position indicator (PPI)

PPIs are radar scans where the antenna performs a complete rotation from 0° to 360° in AZ at a constant EL. Here the EL was set to the exact EL that was determined during the alignment of the external signal source. Data from the PPIs can be used to describe all side lobes including the back lobes of the antenna in the horizontal plane. The data acquired by this scan constitutes the main source for the evaluation in the following section. In total, 5 PPIs were recorded during the measurement, equally spaced between the slices of the large raster scan.

### 165 2.2.4 Range height indicator (RHI)

RHIs are complementary to PPIs: the AZ is fixed to the position of the external signal source, and the EL is scanned from -2° to 90°, constituting the limits of the mechanical system of the radar. The data is used to describe the main and side lobes in the vertical plane. Again, 5 RHIs were performed between the slices of the large raster scan.





**Table 2.** Periods for the precipitation sums and installed LPRs

| Time span | installed LPRs |
|---|---|
| 2022-07-26 to 2022-09-05 (42 days) | old LPR at 357° |
| | 16 mm LPR at 267.5° |
| 2022-09-27 to 2022-10-24 (28 days) | old LPR at 357° |
| | 16 mm LPR at 267.5° |
| | 40 mm LPR at 177.5° |
| | horizontal LPR at 245° (only until 2022-10-07) |

## 2.3 Beam blockage estimation

When the antenna is turning in azimuth, the LPR is locally obstructing parts of the antenna aperture, resulting in a reduced system gain in that direction. The reduced gain can be described as beam blockage, where some of the radiated power is being deflected out of the main lobe on transmit and incoming radiation is partially deflected from reaching the antenna. Two approaches for quantifying this phenomenon are described here.

### 2.3.1 Precipitation sum products

To quantify the effects of beam blockage on weather radar data, one simple approach is to calculate precipitation sums. Over sufficiently long time spans ($\geq 1$ month), it can be expected that the accumulated precipitation field becomes more and more homogeneous around the radar, averaging out local differences caused by variability in precipitation. However, orographic effects will appear more pronounced. Strong negative deviations in precipitation sums that appear as radial spokes in such fields can be attributed to beam blockage.

The MOHP research radar runs the default operational DWD scanning routine most of the time. One of the performed scans is the so-called "precipitation-scan" (PCP), a PPI scan, where the elevation is adapted as a function of azimuth. Hence, the radar beam is guided in such a way that it is always as close to the ground as possible while also being above the horizon. For MOHP this means that the EL is at 0.8° while scanning towards the lowlands in the north and reaches up to 3.5° EL over the Alps in the south. One of these precipitation scans is performed every 5 minutes. The data was recorded over two periods of 185 about a month, each with a different setup of LPRs. Details are summarised in Table 2.

The recorded reflectivity data were transferred into rain rates, using the standard Z-R-relationship used at DWD (Schreiber, 1997; Haase et al., 2000):

$$Z = 256 \cdot R^{1.42}, \tag{1}$$





with Z being a reflectivity in linear units and R the rain rate in $\mathrm{mmh^{-1}}$. The calculated sums are evaluated up to a maximum radius of 30 km around the radar. This ensures that only liquid precipitation is measured, for which the Z-R-relationship is valid.

### 2.3.2 Time scans

During a time scan, the radar antenna is permanently pointed towards the same position, and the recorded rays are defined by time spans. The gathered data can be used to produce robust statistical evaluations under the assumption that the environmental conditions change only minimally during the considered time. In the case presented here, the antenna was pointed directly towards the external signal source. The radar transmitter was turned off and settings regarding range and pulse width are identical to those of the other scans, the PRF was set to 1000 Hz. The duration of one time scan was set to 120 s, producing about 14,000 data points per scan. Three LPR setups were evaluated this way:

- no LPR between weather radar and signal source

- vertical LPR with 16 mm diameter between radar and signal source

- vertical LPR with 40 mm diameter between radar and signal source

The purpose of these scans is to precisely quantify the effect of beam blockage on the propagation path by the LPRs on SNR.

## 3 Results

This section presents the results of the measurements introduced in Section 2. Data from the PPI scans conducted during the antenna measurements are used to quantify the effect of the LPRs on the side lobe level and the sum products. Time scans provide the data for the evaluation of beam blockage and the subsequent underestimation of precipitation amounts, which, as discussed in the introduction, stands as the primary issue with the current LPR setup and prompts this evaluation.

### 3.1 Reflection and side lobes

A LPR scatters the electromagnetic field close to the antenna and partially redirects it off the main beam axis for a given azimuth and elevation, resulting in increased side lobes at the position of the LPR. If there is a target with high reflectivity, such as a thunderstorm appearing in the direction of a strong side lobe, the radiation off the main beam axis can be sufficient to produce a backscatter signal that the radar can detect. However, this signal is recorded as originating from the current pointing angle of the main lobe, resulting in mispositioning. Such a case is presented in Fig. 3. It shows the reflectivity recorded during a thunderstorm at the radar Memmingen (MEM) in Summer 2022. The centre of the thunderstorm (shown in purple) is located in the south west of the radar and reaches over 50 dBZ. The radar uses the old LPRs (see Section 2.1) and one of them is located at an AZ of 203°. Exactly at that position, the weather echo is smeared radially over an AZ span of about 22°, which is highlighted with the red circle.





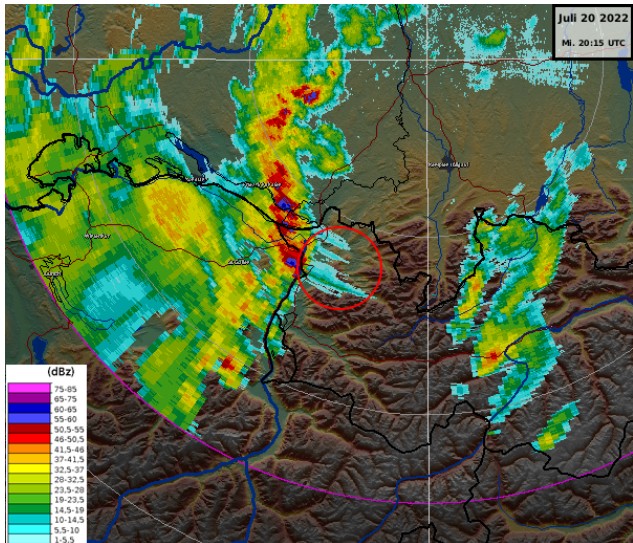

**Figure 3.** Hook echo (red circle) produced by a degraded antenna pattern due to the old LPR at 203° AZ during summer 2022 at the radar MEM.

For the quantification of this effect, SNR measurements from the PPI scans are used. Data is averaged over the recorded range. The maximum recorded SNR is used to normalise the data of each PPI and the position of this maximum is used to centre each PPI on the external signal source. Then, the data is linearly interpolated to an AZ grid of equal spacing with a resolution of 0.05° to account for slight deviations in antenna positioning between the PPIs. Subsequently, the data from the 5 PPIs is averaged into one dataset, resulting in one data point for every 0.05° of AZ, with the maximum SNR set to 0 dB centered at 0° in AZ.

Results from the measurements for the horizontal channel are shown in Fig. 4. As the antenna is moving through the beam of the external signal source, the received SNR will rise and peak when it hits the exact position (0° AZ in Fig. 4) and then decline again. Displayed are SNR values for the measurements without an LPR around the signal source, with one of the three vertical LPRs installed at 267.5° AZ and with the horizontal LPR installed at 245° AZ (relative LPR positions are indicated by gray lines at the top). The black line refers to the antenna pattern without any LPR. It serves as the reference for the antenna pattern results with the investigated LPR options. The most prominent line in blue is for the old LPR. It features increased side lobes over almost all of the shown AZ span. Near the location of the LPR at 17.5°, side lobes are increased by on average up to 20 dB compared to all other measurements. These increased side lobe levels are the resemblance of the earlier showcased 'hook echo' (Fig. 3) in the antenna pattern. The first side lobes, as an important criterion for the assessment of an antenna, are not influenced by the new LPRs and stay close to the reference value of -34.7 and -28.7 dB to the left and right of the main lobe, respectively. This asymmetry has been known since the installation of the antenna. The old LPR led to values of -28.8 dB (left) and -23.7 dB (right). Maximum differences in side lobe level between the old LPR and no LPR reach 40 dB due to small variations in the positioning of the side lobes.

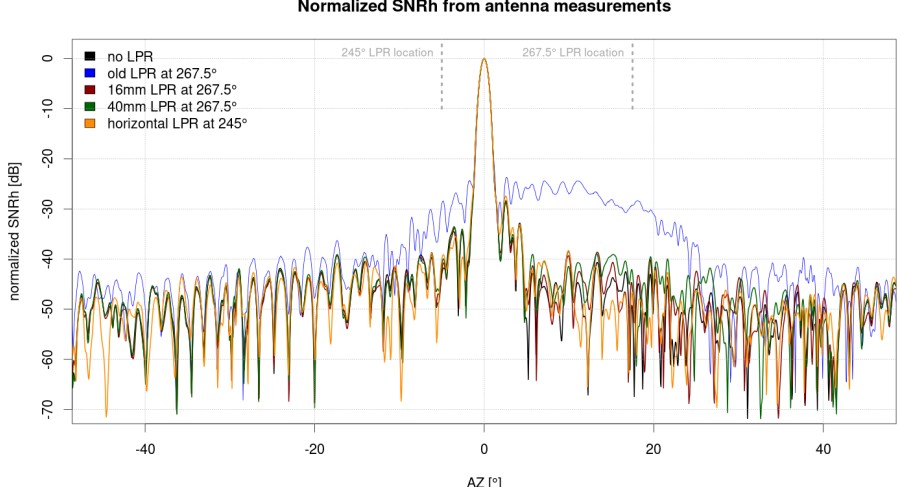

**Figure 4.** Antenna pattern measurement without LPR and four different LPR designs. A radome is present for all measurements. Shown are normalised SNR measurements from PPI scans for the horizontal channel. Grey lines at the top mark the installation locations of the LPRs relative to the peak of the external signal source.

The two vertical LPRs (green and red lines) cause a slight increase in side lobes in close proximity to the rod's position of no more than 5 dB, with the 40 mm diameter rod having a larger impact than the thinner 16 mm rod. The horizontal LPR has no discernible effect on the side lobes at its position other than causing a different shape in the peaks. It is not clear if these differences are caused by the LPR or by having a slightly different setup in the measurement, because it was conducted a few days after the other ones.

Figure 5 shows the data from the vertical channel of the radar. Again, the old LPR has the largest effect on side lobe level. Generally, the signal strength in the side lobes is higher in all measurements compared to the horizontal channel, even with no LPR present. Yet, the relative differences between measurements are more pronounced: in the area of the mounting point of the vertical LPRs, side lobes are clearly increased (for example 6.5 dB difference between no LPR and 16 mm LPR), and their shape is changed, disturbing the typical pattern of minima and maxima that is seen for example at -40° AZ. Apparently, the interaction between the microwave and the LPR is different depending on the polarization of the radiation. In this case, it leads to an increase in the disturbance of the antenna pattern in the vertical polarization plane compared to the horizontal plane. The horizontal LPR again has no discernible effect on the antenna pattern. First side lobes experience a slight increase with the two new vertical LPRs by 0.2 dB (left side) and 0.5 dB (right side).

In summary, the antenna patterns show that all three tested LPRs provide considerable improvement in the overall antenna pattern compared to the old lightning protection solution, with the horizontal rod having the best properties.

So far only the main azimuthal plane of the antenna pattern has been considered. The improvement of the new LPRs on the antenna pattern can also be shown in the 2d plane. Results are shown as an example for the old LPR and the 40 mm LPR for

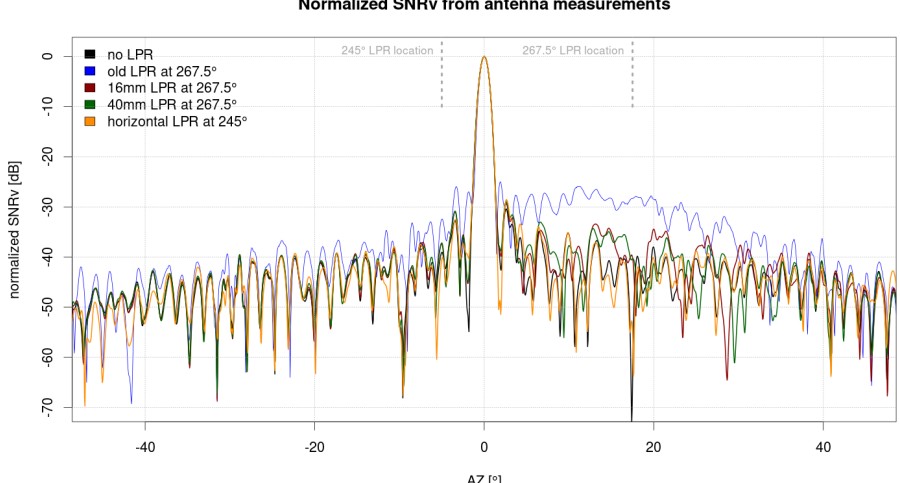

**Figure 5.** As in Fig. 4, but for the vertical channel.

the horizontal channel in Fig. 6 and the vertical channel in Fig. 7. For this analysis, data from all slices of the large raster scans (see Section 2.2) were centered and normalised. Each slice was individually normalised and centered using the nearest small raster scan in time. Subsequently, the normalised slices were combined into one comprehensive data set and interpolated to a common grid.

The red spot at 0° AZ and 0° EL in both panels marks the main beam of the antenna with a beam width of about 1°. The two
areas of about -40 dB going upwards diagonally from there are caused by the antenna struts. When the antenna points above the signal source, some of the incoming field is reflected into the antenna aperture by the struts.

The areas of strong signal reaching out from the signal source to the left and right at 0° EL are the side lobes discussed above. They are clearly stronger, more persistent and fill a larger area in the old LPR case. Compared to the measurement without an LPR (not shown), SNR is increased by up to 15 dB around the LPR location in the 40 mm case. In contrast, in the
case of the old LPR, the signal difference exceeds 30 dB.

### 3.2 Beam blockage

Beam blockage describes the effect that a portion of the transmitted power that is returned from a scatterer is lost on either the transmit or the receive path. The loss is the same in both directions. It is caused by scattering of the electromagnetic wave on an obstacle in such a way that it does either not reach the intended target on the transmit path or the antenna on the recieve
path. For a weather radar, this means that weather with a certain reflectivity will be recorded at a lower dBZ-value behind a LPR than without the obstacle in place. The old LPRs used at the DWD radars are known to cause a loss in reflectivity and subsequently of precipitation rates of up to 30%.





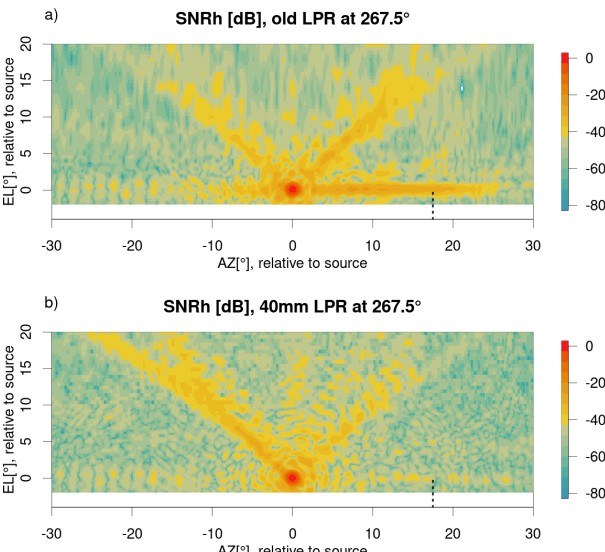

**Figure 6.** SNR (horizontal channel) from the raster scans. a) data from the measurement with the old LPR from 2012, b) data taken with the 40 mm LPR. The vertical black line in the figures marks the position of the LPRs.

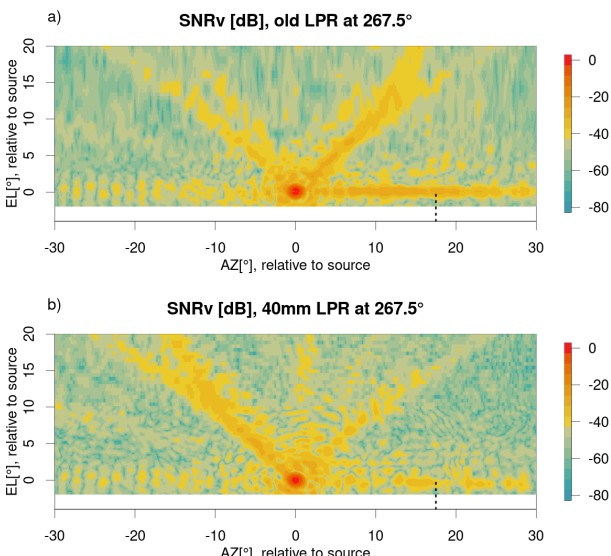

**Figure 7.** As in Fig. 6, but for the vertical channel.

Figure 8 shows the precipitation sum products as they are described in Section 2.3.1 for the time spans and LPR setups listed in Table 2. The black sector between 99° and 111° AZ is caused by safety sector blanking. Despite the summation over 42 days, the August period does not show a homogeneous precipitation field. During this month, rain mostly originates from localised





convection and the southeast of the radar is a known hot-spot for thunderstorms. Still, a cone-shaped area of low precipitation sums is visible in the north, where the old LPR is located. Another distinct beam blockage is apparent at 125° AZ, caused by an old measurement tower with a maximum diameter of about 1 m in 50 m distance from the radar. The comparison between both obstructed areas shows that the beam blockage has approximately the same strength, but the width of the two resulting

cones is distinctly different.

During the October period (bottom panel in Fig. 8), the three features described above (sector blanking, beam blockage by old LPR and measurement tower) are again visible. Additionally, there are several step-like features in 220° to 275° AZ. These are caused by the changes in elevation during the terrain-following precipitation scan. With different EL angles, the most reflective part of the weather (often times the bright band, Klaassen 1988) appears at different distances. These steps were

performed during the August period as well, but are not that clearly visible due to the intrinsic variability in the precipitation field.

In contrast to the August period, in October the three new LPRs were installed at 177.5°, 245° and 167.5° AZ (details in Table 1). At these AZ angles, no sign of any decrease in the precipitation sum is visible. This is confirmed by Fig. 9, where precipitation sums for selected AZ angles are shown for both evaluated periods. The sums shown in Fig. 8 were averaged in range

over 30 km, providing one value per AZ and further smoothing the local variability. At 360° AZ, the strong beam blockage caused by the old LPR is clearly visible: compared to the unobstructed AZ-areas at ±40° AZ, the measured precipitation sum behind the LPR drops by up to 25% in the August period and by ≈15% in October, which again highlights the exact reason for this study. At the angles of the other three LPRs, no such drop in precipitation sum for October is visible, neither as an unique feature for this measurement, nor in comparison to the August measurement without the LPRs installed. Inhomogeneities can

be explained by either local precipitation variability (150°, August) or the change in the elevation angle of the precipitation scan (275°, both periods).

To quantify the beam blockage on the receive path, the time scan measurements (Section 2.3.2) are used. They provide an estimate on how much of the signal from the external source is blocked by the 16 mm and 40 mm LPRs. Panel a) of Fig. 10 shows the absolute SNR for the horizontal (red) and vertical (blue) channels for the measurements with, from left to right,

no LPR, 16 mm LPR and 40 mm LPR. The small width of the boxes including the outliers (≈0.5 dB) prove the robustness of the measurements. Since all three setups were tested within one hour and without any changes in the setup of the external signal source, no change in signal strength due to external factors is expected and all observed differences can be attributed to the LPRs. The difference in signal strength between horizontal and vertical channel is due to the fact that a higher precision is hardly achievable in the manual setup of the external signal source. For both channels, a drop in median SNR from no LPR

to with LPR is visible. To further illustrate, panel b) of Fig. 10 shows the relative differences, where the respective median of the data with no LPR has been subtracted from both channels. Now the effect of beam blockage becomes more clear: In the horizontal channel, the SNR drops by 0.057 and 0.076 dB for the 16 mm and 40 mm LPRs, respectively. The slightly higher drop at the 40 mm rod can be attributed to its larger diameter, providing a larger obstacle for the electromagnetic field. In the vertical channel, the beam blockage is slightly higher. The 16 mm rod lowers the signal by a median of 0.11 dB, the 40 mm



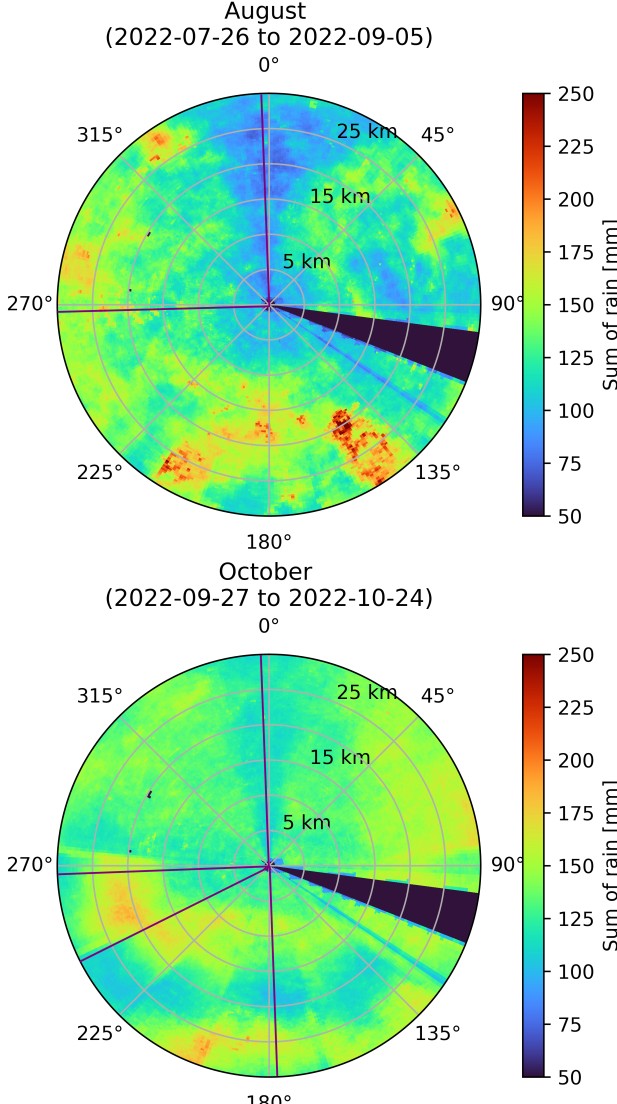

**Figure 8.** Precipitation sums (mm) for the inner 30 km during August (top) and October (bottom). Positions of the installed LPRs are indicated by purple lines. Data originates from terrain following scan with azimuth-dependent elevation changes.

rod by 0.124 dB. This is in accordance with the findings from the antenna patterns, where a stronger influence of the LPRs on the vertical channel is visible as well.

The recorded SNR values can be transformed into reflectivities in dBZ using a range correction, a correction for gas attenuation and a calibration constant. Then, relative changes in dBZ can be calculated: The 16 mm LPR lowers the recorded reflectivity by 0.11% and 0.21% for horizontal and vertical channels, respectively, the values for the 40 mm LPR are 0.14% and 0.24%. The results are doubled to represent the loss on both transmit and receive path and applied to a range of possible

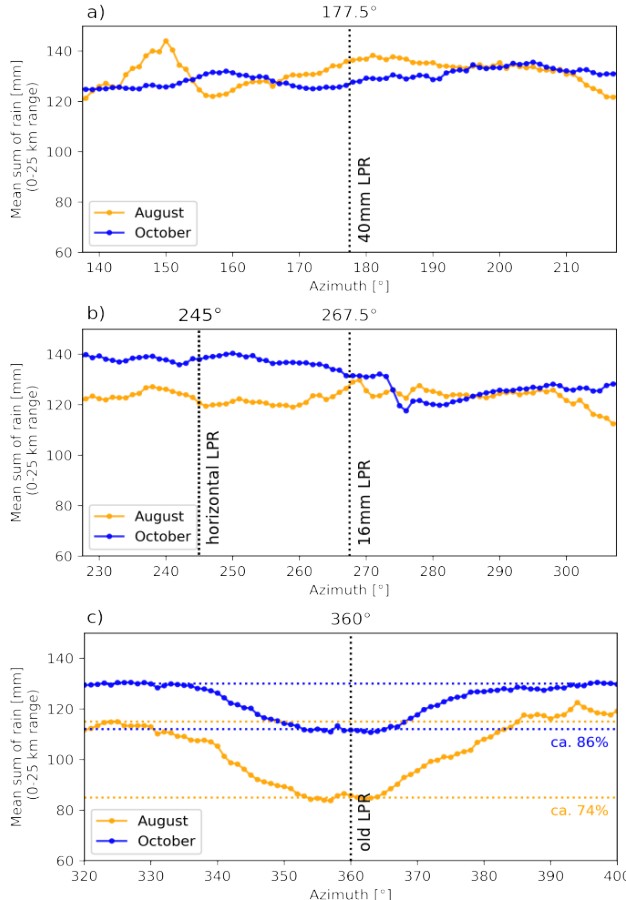

**Figure 9.** Azimuthal averages of precipitation sums shown in Fig. 8 for August (orange) and October (blue). Dotted lines show the positions of the LPRs.

reflectivites. Using the Z-R-relationship in Eq. (1), percentage reductions in the expected rain rates become available: At very high reflectivities over 50 dBZ, losses exceed 1.8% and 2.2% in the horizontal channels of the 16 mm and 40 mm LPRs, respectively, and reach 3.3% and 3.8% in the vertical channels. Usually, such high reflectivities occur only very rarely, and most precipitation originates from events with much lower reflectivity. At a much more frequent value of 25 dBZ, losses are

only 0.9% (16 mm LPR) and 1.1% (40 mm LPR) for the horizontal channels, and 1.7% and 1.9% for the vertical channels. In fact, the mean of all reflectivity data points that were used for the rain sums shown for the month of August in Fig. 8 is 4.8 dBZ. The loss in rain amount there is 0.39% in the worst case (40 mm LPR, vertical channel). This indicates why no reductions are visible at the locations of the new LPRs in Fig. 8 and 9.

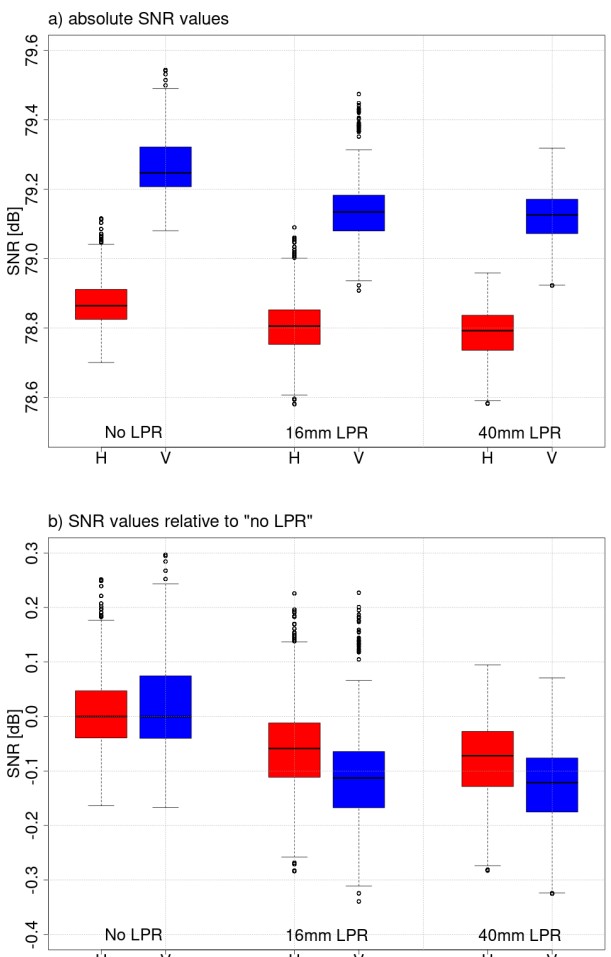

**Figure 10.** SNR from the time scans described in Sec. 2.3.2. a) absolute measured values, b) values relative to those of the "No LPR" measurent.

## 4 Discussion and conclusion

The results from Section 3 show that the evaluated LPRs do have an effect on data quality when compared to the conditions without any rods, but it is minimal compared to the old LPR. Therefore all three setups can be recommended for future lightning protection at DWD radar sites.

Antenna diagrams show that the horizontal LPR has the smallest effect on the measurements, due to the fact that very little of it covers the aperture of the antenna during scans. In fact, this setup seems to have the lowest minima in between the side 330  lobes and at some areas of the diagram.



There are many factors that might introduce slight changes in the measurements that are not caused by the LPRs. These include the repeated manual setup of the external signal source, slight differences in the positioning of both antennas, differences in the absolute signal power between measurements and the atmospheric effects on beam propagation and attenuation. Despite the short time required for one measurement (two hours), the campaign had to take place over the span of two weeks, including

changes in weather and the repeated setup of the external signal source.

The summation of rain rates is a simple method of quantifying the effects of beam blockage. The possible severity of this effect was shown with the old LPR, behind which the amount of rain was substantially underestimated. Still, the method has its shortcomings. Even over a month long period, precipitation fields can be so variable that no homogeneous sums are produced. Ideally, the evaluation is made over at least one year in order to eliminate seasonal differences. As shown, the percentage

underestimation of precipitation behind the rod is not always the same. This is potentially caused by differences in the size of rain drops. During August, most of the rain comes from convective events with large drops, while in October, stratiform rain increases in frequency. According to the radar equation, the measured reflectivity is a function of the 6th power of drop size (e.g. Doviak and Zrnic, 1993). If therefore the same percentage amount of power is blocked from a strong signal, this results in a larger loss of recorded rain and therefore reflectivity than if the blocking happens to a weak signal

The three tested new LPRs did not show any decrease in precipitation sum at their location during the recorded month. As noted above, it can not be ruled out that an effect might become visible over substantially longer evaluation periods of one or several years. As discussed, the loss in recorded rain amounts caused by the two evaluated vertical LPRs can reach up to 3.8% during very high intensity precipitation events, but do not exceed 1% loss during most of the time.

From a data centered point of view, the horizontal LPRs have the smallest effect on data quality and should therefore be

recommended as a lightning protection concept if one has the freedom to design it and the tower it's placed on from scratch. By varying the number of rods and their length, different lightning protection levels can be realised in accordance with the regulations in Germany. In the case of the DWD radar network, the new concept has to be adapted to the existing infrastructure of towers and their surroundings and radars. All evaluated LPRs are expected to be subject to ice formation under the right meteorological conditions. In case of the horizontal rods, the ice might break and fall down around the tower, causing a

potential hazard to people and infrastructure on the surrounding ground. Additionally, accumulation of ice might lead to a structural failure of the rods and make them break. This concept is therefore not readily applicable to existing sites with their individual requirements and available infrastructure.

For the 16 mm LPR, calculations of potential ice formation showed that the diameter is too small for structural integrity at MOHP (a mountain site), meaning that the rods might break if a lot of ice is accumulating around them and they are hit by

strong winds. This leaves the LPRs with 40 mm diameter as the only structurally serviceable lightning protection concept and is therefore recommended for retrofitting the weather radar network of the German meteorological service.

*Code and data availability.* The recorded data from the antenna measurements and the R-code for the evaluation are available upon request from the authors.



*Author contributions.* CH helped design the study, wrote large parts of the paper, developed the scanning routines and evaluated and plotted
the antenna measurement data. MS developed the scanning routines, supported the data evaluation and wrote parts of the paper. AB developed
the rain sum evaluation and collected, evaluated and plotted the data. MF helped design the study, wrote parts of the paper and supported the
operation of the radar during the study. JP supported the operation of the radar and the external signal source during the study. BL supported
the operation of the radar and the external signal source during the study. BR helped design the study, wrote parts of the paper and supported
the operation of the radar during the study.

*Competing interests.* The authors declare no competing interests.

*Acknowledgements.* The authors want to acknowledge Kai Reymers for the installation of LPR options, Karl Zirngibl for supporting the
operation of the external signal source and Tabea Wilke for providing the precipitation sum algorithms. All named colleagues are at DWD.





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
