# Peer review of "Evaluation of the effects of different lightning protection rods on the data quality of C-Band weather radars"

_Atmospheric Measurement Techniques, 2024_

## Author Response (AR1)

**Structure:**

- Reviewers comments (black)
- Detailed answer (blue)
- Line numbers where to find changes, referring to the revised manuscript (blue)

**Comments RC1:**

The paper is structured well. In the introductory section the theoretical background, the current status of lightning protection at DWD as well as three possible replacements are highlighted shortly. The measurement setup and the measurements itself are described in detail. Two approaches to determine beam blockage caused by LPRs are presented. The results section focus on reflection, side lobes and beamblockage with detailed explanations on the basis of several images/graphics. Results are shortly discussed and the reader can follow the conclusions drawn.

**General comments:**

Under 2.2 Antenna measurements ZDR, PhiDP and RhoHV were listed as recorded radar moments, but not handled throughout the paper again. I've expected some examples or results concerning those radar moments with respect to the different LPR setups, but at least some mention in the conclusions.

The evaluation of the polarimetric moments provided no additional information. Their values are changed in the same places where we find changes in SNR. The variability in the polarimetric values introduced by the LPRs is within the limits seen in the measurement with no LPR; it is just the shape that is different. We added a paragraph to Section 3.1 with this information. You can find figures for the Dual-Pol moments in Frech, M., Lange, B., Mammen, T., Seltmann, J., Morehead, C., and Rowan, J.: Influence of a radome on antenna performance, journal of atmospheric and oceanic technology, 30, 313–324, 2013.

- Line 280-284 added.

In 2.2.4 RHI measurements were taken, is there a reason they haven't been taken into account in the results?

The RHIs were evaluated in the same way the PPIs were. We found no changes caused by the different LPRs. We added a sentence in the introduction to Chapter 3 noting this.

- Line 216ff

A tabular overview of the results would be nice for quicker evaluation.

Thank you for this suggestion. We fully agree and have therefore added a short table with the key findings at the end of the Discussion and Conclusion chapter.

- Added Table 3
- Line 382f: added reference to Table 3

**Specific comments:**

2.1/2.2  Although it is all well described, a graphical top view of the whole setup would be nice. Sketches of the different measurements (eg source raster scans, large raster scan) would also help getting a better insight into the setup.

The relative positioning of the installation locations of the LPRs to each other can be derived from the lower panel in Figure 8. We added this info in Section 2.3.1.

We also added a Figure showing how the different conducted scans are positioned with respect to each other and to the LPRs and substituted the photos of the rods with a sketch that shows the dimensions of the rods more clearly.

- New Fig. 1

- New Fig. 2

- Lines 142f, 154, 163f, 170f, 175f: reference to new Fig. 2

- Line 193f: Reference to Fig. 8

2.2.1 Row 148 Please account for/describe pointing accuracy.

We have added the following sentence here: „This is well within the achievable angle resolution of the radar which is plus / minus 0.007° according to the angle tags in the data." This shows that we can reasonably set our data resolution to 0.05° and get independent rays. A note about the absolute pointing accuracy is given at your second comment about the pointing accuracy.

- Line 151f

2.3.1 Row 189-191 Please explain that seemingly arbitrary range of 30km. We can assume that this is well below the ML, but a hard number of max beam height at 30km range would be nice.

We inserted the height of the measurements at 30km range and added that the limitation is used in order to stay below the melting layer.

- Line 199f: Added the information in brackets and „below the melting layer"

3.1 Row 220/221 In which range are the mentioned slight deviations in antenna positioning? See first comment on pointing accuracy.

We have added a sentence describing that the interpolation is necessary in order to avoid having to deal with the slight variations in angle tags of about 0.007° when averaging the sweeps. The absolute pointing accuracy of better than 0.1° (see source below) does not play a role, because every sweep is normalised by its own maximum SNR value. Therefore, the signal source is at 0° AZ always.

Frech, Michael, Theodor Mammen, and Bertram Lange. "Pointing accuracy of an operational polarimetric weather radar." *Remote Sensing* 11.9 (2019): 1115.

- Line 233-235: Added an explanatory sentence

Row 234 Is there an explanation for this asymmetry?

During the first antenna measurements in 2013 this was attributed to a possible lateral feed misalignment. We have added this information at the indicated location.

- Line 253f: added explanation and source.

3.2 Fig6/Fig7 Although one can follow the explanation of the figures very well, you might consider creating additional difference plots to make differences in the results more visible. Also adding some markers into the images to explain several features might help to quickly comprehend, especially as the images might be located on another page disconnected from the describing text.

Thank you for this suggestion, we fully agree that this would help the figures to stand for themselves. We added panels showing the differences between measurements with and without lightning protection rods to show influenced areas more clearly. We also added markers for the signal source, the struts and the influenced side lobe area in the respective first panel of each figure.

- Fig. 6 and 7: added two more panels to each, added markers to panel a
- Fig. 6: added info about new panels to caption.
- Line 273: clarified reference to figures 6 and 7 with a) and b)
- Line 278: removed „(not shown)" and added reference to panels c) and d)

**Technical corrections:**

row 118 alteratively -> alternatively

The typo has been fixed.

- Line 119

row 128/129 The PRF of the radar is set to ~is at~ 3000 Hz and the pulse width to ~to~ 0.4 µs.

The sentence has been completely rewritten due to the comment of another reviewer.

- Line 130ff

row 316 reflectivites -> reflectivities

The typo has been fixed.

- Line 336

row 344 period missing at end of sentence

The missing period has been found and was inserted at its proper location.

- Line 363

row 346 can not -> cannot  (that's debatable, though)

We followed your suggestion and removed the space.

- Line 367

**Comments RC2:**

The manuscript describes the lightning protection of weather radars operated by German Meteorological Service. The current system has disadvantages concerning data quality. A new design of lightning protection rods (LPR) shows considerable reduction of beam blocking.

The manuscript is well prepared and describes detailed and comprehensive. All necessary information to follow the analysis of the measurements with different LPRs is provided. The manuscript can be accepted for publication after minor changes.

**Comments:**

Line 34: hail-spikes are in radial direction; the example shows spurious echoes in azimuthal direction.

Thank you for this note. We changed the sentence to say the following: „In an actual radar image of a thunderstorm this appears as an echo that has some resemblance to a so-called hail spike, but extends in the azimuthal instead of the radial direction.“

- Line 34f

Line 63: specify, that these measurements are only done for the Hohenpeißenberg radar

We added a few words, specifying the location of the measurements.

- Line 64f

Figure 1 and 2: the photos are fine, but a sketch showing all four types of LPRs would be helpful

Thank you for this suggestion! We agree that a schematic would be more meaningful. We substituted the two photos with a sketch showing all four LPRs in comparison.

- Line 74: removed reference to former Fig. 1
- Line 80: removed reference to former Fig. 2, added reference to new Fig. 1, changed reference to Tab. 1
- Added new Fig. 1
- Line 99: removed reference to fromer Fig. 2, substituted by a explanation.

Line 129: As I understand that these measurements are done while the transmitter is off (line 92). It might be better to write, that the bandwidth and bin resolution is set to the values which are used in case of the transmitter pulse width is 0.4 µs.

The radar uses software and hardware limits to ensure that the maximum duty cycle is not exceeded. So setting a PRF of 3000Hz to achieve a high sampling rate is only possible when selecting our shortest available pulse width of 0.4µs. This of course means that the bandwidth and bin resolution of this pulse width also apply to the measurement with the transmitter turned off. We clarified this in the manuscript.

- Line 130ff

Line 187: a and b in Eq (1) originate from Aniol et al., 1980: Über kleinräumige und zeitliche Variationen der Niederschlagsintensität. Meteorol. Rdsch., 33, 50-56.

Thank you, we changed the source to the original publication.

- Line 195f

Line 189: … reflectivity in linear units "(mm^6 / m^3)" and ...

We added the units for the linear reflectivity.

- Line 198

Figure 3: a hook echo is often related to tornadoes, it resembles a hook, which is not the case in this example. I would call this a fake or apparent echo

We initially called the phenomenon hook echo, since it was the closest thing we know from radar meteorology that resembled what we saw. Yet, we agree that this might cause confusion and changed the name to „spurious echo".

- Annotation to Fig. 3

Line 220: ... linearly interpolation ... does this refer to spatial interpolation? Or SNR in linear units? Or both, please specify.

The data was interpolated spatially using a bilinear interpolation on the logarithmic SNR values. We clarified this in the manuscript.

- Line 231f

Line 233: see comment to Fig. 3 caption

see above.

- Line 244: changed „hook" to „spurious"

Line 244: is there any explanation to the different strength of the side lobe power for H and V polarization?

We have no explanation for this difference in the vertical and horizontal polarization plane. It is a characteristic of this specific feed and antenna combination that has been known since the factory acceptance tests.

- No changes made.

Line 276: … convection and the "region" southeast of the radar is …

We changed the part to „the region in the southeast".

- Line 295

Line 285: I think in August the melting layer can be higher than October and also convective precipitation with no pronounced melting layer can be more dominant

We added both points as an explanation for the higher variability in the precipitation field during that period.

- Line 305f

Line 317 and 319: losses in rain rate

„in rain rate" was added to both instances.

- Lines 337, 339f

Line 340 to 342: this is a nice trial to explain the differences in convective and stratiform precipitation. It's easier to say that this is related to the nonlinear behavior (i.e. exponent is not 1) of the Z-R relation.

We removed the sentence „This is potentially caused by differences in the size of rain drops." and added „The nonlinear behaviour of the Z-R-relationship that results in exponentially higher rain rates for high reflectivities intensifies this effect." to the end of the paragraph. In our opinion, the differences in in the measured loss of rainrate stems from the nonlinear behaviour of both the radar equation and the Z-R-relationship.

- Lines 360, 364f

Line 350: ... the tower it is placed on ... would be easier to read for me, but I'm not native English

We agree, it sounds better that way.

- Line 371